# Genomic Characterization of Carbapenemase-Producing Enterobacteriaceae from Clinical and Epidemiological Human Samples

**DOI:** 10.3390/antibiotics14010042

**Published:** 2025-01-06

**Authors:** Alexander Tristancho-Baró, Laura Eva Franco-Fobe, Monica Pilar Ariza, Ana Milagro, Ana Isabel López-Calleja, Blanca Fortuño, Concepción López, Miriam Latorre-Millán, Laura Clusa, Rosa Martínez, Carmen Torres, Antonio Rezusta

**Affiliations:** 1Clinical Microbiology Laboratory, Miguel Servet University Hospital, 50009 Zaragoza, Spain; lefranco@salud.aragon.es (L.E.F.-F.); mparizas@salud.aragon.es (M.P.A.); amilagro@salud.aragon.es (A.M.); ailopezcal@salud.aragon.es (A.I.L.-C.); bmfortuno@salud.aragon.es (B.F.); clopezgo@salud.aragon.es (C.L.); arezusta@salud.aragon.es (A.R.); 2Research Group on Difficult to Diagnose and Treat Infections, Institute for Health Research Aragon, Miguel Servet University Hospital, 50009 Zaragoza, Spain; mlatorre@iisaragon.es (M.L.-M.); lclusa@iisaragon.es (L.C.); 3Infectious Diseases Department, Miguel Servet University Hospital, 50009 Zaragoza, Spain; rmartineza@salud.aragon.es; 4Area of Biochemistry and Molecular Biology, One Health-UR Research Group, University of La Rioja, 26006 Logroño, Spain; carmen.torres@unirioja.es

**Keywords:** carbapenem-resistant enterobacteriaceae, whole-genome sequencing, drug resistance, microbial, computational biology

## Abstract

**Background/Objectives**: Infections caused by multidrug-resistant (MDR)bacteria pose a significant public health threat by worsening patient outcomes, contributing to hospital outbreaks, and increasing health and economic burdens. Advanced genomic tools enhance the detection of resistance genes, virulence factors, and high-risk clones, thus improving the management of MDR infections. In the Autonomous Community of Aragon, the diversity and incidence of carbapenemase-producing Enterobacteriaceae (CPE) have increased during the last years. This study analyses CPE trends at a tertiary hospital in Spain from 2021 to 2023, aiming to optimize personalized medicine. **Methods**: CPE isolates were the first isolate per patient, year, species, and carbapenemase from January 2021 to December 2023. Additional metadata were collected from the laboratory’s information system. Antibiotic susceptibility testing was performed by broth microdilution. Whole-genome sequencing (WGS) was performed using Illumina short reads. De novo assembly was used to generate draft genomes in order to determine their complete taxonomic classification, resistome, plasmidome, sequence type (ST), core–genome multilocus sequence typing (cgMLST), and phylogenetic relationships using a suite of bioinformatics tools and in-house scripts. **Results**: Between 2021 and 2023, 0.4% out of 38,145 Enterobacteriaceae isolates were CPE. The CPE rate tripled in 2022 and doubled again in 2023. The most common species was *Klebsiella pneumoniae* (51.8%) and the most common carbapenemase was *bla_OXA-48_*. WGS revealed concordant species identification and the carbapenemase distribution in detail. Resistance rates to critical antibiotics, such as carbapenems, were variable, but in most cases were above 70%. Genetic diversity was observed in WGS and phylogenetic analyses, with plasmids often mediating carbapenemase dissemination. **Conclusions**: The increasing rate of CPE in healthcare settings highlights a critical public health challenge, with limited treatment options. Genomic characterization is essential to understanding resistance mechanisms, aiding therapy, limiting outbreaks, and improving precision medicine.

## 1. Introduction

Infections caused by multidrug-resistant (MDR) microorganisms represent a significant global public health issue [1,2]. The isolation of MDR microorganisms, including carbapenemase-producing enterobacteria, from clinical or epidemiological samples often correlates with poorer prognoses and increased morbidity, particularly when other virulence factors are present [3]. Furthermore, many of these microorganisms have the ability to spread within communities and hospitals, leading to outbreaks and epidemics [4,5]. The accurate characterization of resistance determinants, virulence factors, mobile genetic elements, and the clonal relationships among isolates can be achieved using high-throughput sequencing and bioinformatics tools. This enables the detection and tracking of high-risk clones, phenotypic correlation with antimicrobial susceptibility testing, and control of hospital outbreaks. These results complement those obtained by traditional methods and have proven useful in the management of infections and outbreaks caused by MDR microorganisms worldwide, leading to their increasingly widespread use in clinical microbiology and public health [6,7].

Carbapenems are a class of broad-spectrum β-lactam antibiotics that exert their bactericidal effect by inhibiting penicillin-binding proteins, thereby disrupting bacterial cell wall synthesis. These agents are frequently used to treat Gram-negative bacteria resistant to other therapeutic options or for the empirical management of critically ill patients in settings with a high prevalence of MDR pathogens. The most commonly used carbapenems in clinical practice are ertapenem, imipenem, and meropenem [8].

Carbapenemases are enzymes capable of hydrolysing the β-lactam ring, thereby inactivating a wide range of β-lactam antibiotics, including carbapenems. According to the Ambler classification, carbapenemases are categorized into four molecular classes, with the most clinically relevant being Class A (e.g., KPC), Class B (e.g., NDM, VIM, IMP), and Class D (e.g., OXA-48-like). Notably, Classes A and D utilize a serine residue in their active site to promote enzymatic activity, while Class B enzymes depend on metal ions, typically zinc, for their function [9].

In 2017, the World Health Organization (WHO) published a priority list of pathogens to guide the research and development of new antibiotics, categorizing carbapenemase-producing Enterobacteriaceae (CPE) as a critical priority [10]. CPE cause difficult-to-treat infections, increase morbidity and mortality, and promote high transmissibility, thus necessitating their active surveillance. Additionally, their prevalence is high in certain countries, increasing the likelihood of transmission, particularly in hospital settings [11].

In 2021, as part of the National Plan against Antibiotic Resistance (PRAN) in Spain, the Network of Laboratories for the Surveillance of Resistant Microorganisms (RedLabRA) was created, with the primary objective of achieving comprehensive and high-quality microbiological diagnostics [12]. This network integrates genomic sequencing in all cases of infection or colonization by antibiotic-resistant microorganisms under surveillance in the National Health System (currently including *Escherichia coli, Klebsiella pneumoniae*, and *Enterobacter cloacae* complex-producing carbapenemases).

In the Autonomous Community of Aragón, Spain, the isolation of CPE has been sporadic and was previously limited to OXA-48-producing *K. pneumoniae*, leading to hospital outbreaks that were identified and contained [13]. However, in recent years, both the number and diversity of bacterial species and carbapenemases have steadily increased, creating new paradigms for their surveillance, detection, treatment, and interdisciplinary evaluation. These findings may be related to the aftermath of the COVID-19 pandemic and the forced displacement caused by the war in Ukraine [14,15], which has altered our hospital’s medical care and referral system of military and civilian incidents. In this sense, the Miguel Servet University Hospital (HUMS) has deemed it essential to monitor CPE and vancomycin-resistant *Enterococcus* due to their increasing incidence and the impact of their spread in the community.

The integration of genomic data analysis into clinical microbiology, with a translational perspective, could yield results that add significant value to the care of patients with MDR microbial infections and advance the pursuit of personalized medicine [16].

In the ever-evolving landscape of global healthcare, the rise of MDR has become a pressing concern, posing a significant threat to public health and challenging the efficacy of conventional antimicrobial treatments. This research delves into the prevalence and characteristics of CPE in a tertiary hospital in Spain over the course of a three-year period, providing a comprehensive analysis of its typing and antimicrobial resistance determinants.

## 2. Results

### 2.1. Case Selection and Species Identification

From January 2021 to December 2023, a total of 38,145 *Enterobacteriaceae* isolates were recovered in the clinical microbiology laboratory at HUMS. Approximately 97% of these (n = 36,956 isolates) were obtained from clinical samples, with *Escherichia coli* (60.5%), *Klebsiella pneumoniae* (17.1%), *Proteus mirabilis* (6%), and *Enterobacter cloacae* complex (4.1%) being the most prevalent species. During this period, 112 CPE were identified, representing 0.4% of the total isolates. Additionally, the distribution of both the total number of isolates and CPE was not uniform throughout the study period, as there was a consistent increase along the years in the total number of enterobacteria and CPE. Specifically, the total number of enterobacteria increased by 18.1% in 2022 compared to 2021 and by 19.3% in 2023 compared to the previous year. However, the increase in CPE was greater than what could be explained solely by the increase in the total number of enterobacteria isolates, almost tripling in 2022 compared to 2021, representing 0.2% (*p* value < 0.01) of the total isolates, and doubling again in 2023, representing 0.5% (*p* value < 0.01) of total enterobacteria. Total isolations per species and year and the presence of carbapenemase are presented in Table 1.

The criteria for selecting CPE isolates were the first isolate per patient, year, species, and type of carbapenemase. Consequently, the 112 CPE isolates corresponded to 93 unique patients, of whom 74.1% were male. Patients’ ages ranged from 0.6 to 90.7 years, with a median age of 51.2 years. Epidemiological samples, mainly rectal and perianal swabs, accounted for 68% of the isolates, while the remaining 32% were from clinical samples. Among the clinical samples, approximately 50% were urine, 25% were from surgical wounds, and the remaining 25% were from abscesses, tracheal exudates, other wound types, ascitic fluid, and blood. Most isolates were obtained from hospitalized patients, with 17% from patients admitted to an intensive care unit. Moreover, 66% of the isolates were from HUMS, and the remaining 34% were from other hospitals for which HUMS serves as a reference centre. Overall, 110 of the 112 CPE had undergone antibiotic susceptibility testing, and 91 were available for sequencing (81%). For the remaining 21 strains, there was no viable sample with which to re-culture for sequencing.

*K. pneumoniae* was the most common CPE species, accounting for 51.8% of the isolates, followed by *E. coli* (18.8%), *Citrobacter* spp. (16.1%), *Enterobacter cloacae* complex (7.1%), *Providencia stuartii* (2.7%), *Klebsiella oxytoca* (1.8%), *Proteus mirabilis* (1%), and *Serratia marcescens* (1%). This species distribution was comparable to that in the subset of CPE isolates that underwent WGS, consisting of *K. pneumoniae* (n = 53), *E. coli* (n = 21), *E. cloacae* (n = 7), *Citrobacter* spp. (n = 7), *P. stuartii* (n = 3), *K. oxytoca* (n = 2), and *P. mirabilis* (n = 1).

WGS yielded concordant species identification compared to matrix-assisted laser desorption ionization time of flight (MALDI-TOF), while also providing enhanced resolution for distinguishing closely related species such as *K. oxytoca* and *K. michiganensis* or within complexes such as in the *Citrobacter freundii* complex, in which two isolates were reclassified as *Citrobacter cronae* and *Citrobacter portucalensis*.

### 2.2. Antibiotic Susceptibility

The antibiotic susceptibility analysis showed high rates of resistance among CPE isolates. Resistance to cephalosporins such as ceftazidime and cefepime was observed in 77.5% and 84.3% of isolates, respectively. For carbapenems, resistance to ertapenem, imipenem, and meropenem reached 92.1%, 76.4%, and 68.5%, respectively. Approximately three-quarters of the isolates were resistant to the fluoroquinolones ciprofloxacin and levofloxacin. Aminoglycoside resistance ranged from 52.8% for gentamicin to 64% for amikacin and 75.3% for tobramycin. Resistance to the “last-resort” antibiotics fosfomycin, tigecycline, and colistin was 39.3%, 33.3%, and 10.6%, respectively. Additionally, the new-generation antibiotics ceftolozane–tazobactam, ceftazidime–avibactam, and cefiderocol showed non-susceptibility rates of 77.5%, 41.6%, and 31.7%, respectively. These resistance patterns were analyzed considering the intrinsic resistance profiles of the *Enterobacteriaceae* species evaluated, such as the presence of chromosomal AmpC in some of them. The fosfomycin results were limited to *E. coli* and *Klebsiella* spp., and the tigecycline cutoff was based on the *E. coli* interpretation. A detailed description of the antibiotic susceptibility pattern by the carbapenemase type in each isolate is shown in Figure 1.

### 2.3. Resistome

The resistome annotation results encompassed all resistance mechanisms detected in the genomes. However, the analysis only focused on those mechanisms involved in the direct modification of the antibiotic molecule or its target site of action, excluding resistance mechanisms related to cell membrane permeability, active efflux, and the lipopolysaccharide profile.

In total, 101 carbapenemase-encoding genes were identified from the 91 genomes sequenced. The most common carbapenemase found was OXA-48-like, represented by 50 isolates with different alleles of the *bla*_OXA_ gene, i.e., *bla*_OXA-48_ (38 isolates), *bla*_OXA-181_ (5 isolates), *bla*_OXA-244_ (5 isolates), and *bla*_OXA-484_ (2 isolates). This was followed by NDM, with 2 alleles present, namely, *bla*_NDM-1_ (15 isolates) and *bla*_NDM-5_ (9 isolates). The third most common was the VIM-type carbapenemase, mainly *bla*_VIM-1_ (13 isolates). The least frequent was the class A KPC carbapenemase, with a similar distribution observed between *bla*_KPC-2_ (7 isolates) and *bla*_KPC-3_ (6 isolates) alleles. Notably, 10 strains were found to harbour dual carbapenemase genes, comprising *bla*_NDM-1_ + *bla*_OXA-48_ (7 isolates), *bla*_NDM-1_ + *bla*_OXA-244_ (2 isolates), and *bla*_NDM-1_ + *bla*_KPC-2_ (1 isolate).

Regarding other β-lactamases, extended-spectrum β-lactamase genes were detected in 47.3% of the isolates, with the CTX-M-type being the most common, including *bla*_CTX-M-15_, *bla*_CTX-M-9_, and *bla*_CTX-M-55_ alleles. Interestingly, the gene *bla*_VEB-6_ was detected in *P. mirabilis.* No mutations were observed at positions 238 or 179 in the *bla_SHV_* gene, phenomena which confer resistance to third-generation cephalosporins [17]. Because of their epidemiological importance, the presence of class C β-lactamase genes in the genomes of *E. coli* and *K. pneumoniae* was evaluated, with the *bla*_EC_ gene being excluded due to its potential chromosomal nature in *E. coli.* Several alleles of the *bla*_CMY_ gene were detected in seven *E. coli* strains, and *bla_DHA-1_* was found in one *K. pneumoniae* strain. Other β-lactamases detected in the isolates, mainly class A and D with lower hydrolytic profiles, included multiple alleles of *bla*_SHV_, *bla*_TEM_, and *bla_OXA_* genes. In particular, *bla*_LEN-16_ was associated with *bla*_VIM-24_ and *bla*_CTX-M-9_ in one isolate of *Klebsiella variicola*. Only three strains exhibited a carbapenemase gene as the sole β-lactamase in their genome. This included two *Providencia stuartii bla*_NDM-5_ and one *Citrobacter amalonaticus bla*_VIM-1_. Table 2 shows the types of carbapenemases, arranged by species.

Table 2 shows the distribution of carbapenemases according to the host microorganism. Total numbers of carbapenemases and Enterobacteriaceae are showed in the last column and row, respectively.

Regarding fluoroquinolone resistance, 82.4% of the strains harboured genetic resistance determinants, including point mutations in the *gyrA*, *gyrB*, or *parC* genes, as well as the presence of the *qnr* gene. They mainly exhibited multiple alleles of *qnrB*, and lower frequencies of *qnrA*, *qnrD*, and *qnrS*. The concurrent presence of at least one mutation in *gyrA* and *parC*, observed in 55 isolates, conferred phenotypic resistance in 100% of the cases. In contrast, the isolated presence of the *qnr* gene in 13 isolates resulted in variable levels of resistance and discrepant phenotypes, with some strains being ciprofloxacin-resistant but levofloxacin-sensitive. Additionally, 6 isolates had mutations in a single gene, either *gyrA* or *gyrB*, while 29 isolates had mutations in *gyrA/gyrB*, *parC*, and the simultaneous presence of *qnr*. The most common amino acid variant in GyrA was the substitution of serine for isoleucine at position 83, accounting for 43.9% of the cases, with less frequent changes to phenylalanine, leucine, and tyrosine at the same position, as well as other substitutions at position 87. Notably, the D87G substitution exhibited a fluoroquinolone-resistant phenotype, even in the absence of other resistance determinants. Substitutions in GyrB involved the change of aspartate with glutamate and alanine with serine at positions 463 and 466, respectively, the former only being found in the absence of *gyrA* mutations and only conferring a resistant phenotype when combined with the presence of the *qnr* gene. All substitutions in ParC corresponded to a change from serine to isoleucine at position 80, and no strains presented isolated mutations in the *parC* gene.

Genes encoding aminoglycoside-modifying enzymes were detected in approximately 90% of the isolates, with acetyltransferases being the most common type, followed by nucleotidyltransferases and phosphotransferases. More than one of these enzyme-coding genes were found in two-thirds of the isolates, and 30% harboured three or more. The presence of the gene encoding AAC6′-Ib, an enzyme of epidemiological and clinical importance due to its role in amikacin inactivation [18], was specifically investigated and detected in 56 isolates, 50% of which exhibited in vitro resistance to amikacin. The high diversity in the presence and combinations of aminoglycoside-modifying enzymes made it challenging to correlate with specific in vitro susceptibility patterns to gentamicin, tobramycin, and amikacin, which were variable.

### 2.4. Typing

Multilocus sequence typing analysis revealed the presence of predominant sequence types among *K. pneumoniae* isolates, while the less common species exhibited greater genetic diversity. The most prevalent *K. pneumoniae* sequence type was ST147, followed by ST307 and ST395. Additionally, two isolates of the hypervirulent *K. pneumoniae* sequence type ST23 [19] were identified, which also harboured dual carbapenemase genes. In the case of *E. coli*, according to the Achtman MLST scheme [20], only two sequence types, ST2659 and ST167, were detected in two or more isolates, with four and two isolates, respectively. Interestingly, one *K. michiganensis* isolate showed a potential new sequence type, characterized by a G159T transversion in the *mdh* gene, representing a previously undescribed allele [21].

The cgMLST analysis of the *K. pneumoniae* isolates, using a more relaxed threshold of 15 different alleles to distinguish individual clusters given the temporal separation of the isolates [22,23,24], revealed a high degree of genetic diversity (Figure 2). The analysis identified nine clusters containing at least two isolates, but only two of these clusters included four or more isolates. Notably, there was no clear clonal relationship between isolates expressing the same carbapenemase gene, with the exception of the *bla*_OXA-181_ variant, where the four identified strains showed a close genetic relationship. Interestingly, four of the strains harbouring the dual carbapenemase genes *bla_NDM-1_* and *bla_OXA-48_* were grouped within the largest cluster, which was predominantly composed of the ST147 sequence type. In contrast, the *E. coli* analysis, using the same clustering threshold, identified only two clusters that directly corresponded to the observed sequence types shown in Figure 3. Importantly, these *E. coli* clusters exhibited a clear association between the isolates and the specific carbapenemase genes, with the main cluster carrying *bla*_NDM-5_ and the secondary cluster carrying the dual carbapenemase genes *bla*_NDM-1_ and *bla*_OXA-244_. Notably, 2 isolates harbouring *bla*_OXA-484_ were separated only by 35 alleles in the cgMLST and 1 of them was classified as a possible new ST due to a mismatch in the *adk* gene, requiring confirmation by resequencing; the closest match was ST1722.

The phylogenetic analysis of the *K. pneumoniae* isolates revealed a distribution similar to the minimum spanning tree generated from cgMLST (Figure 4 and Figure 5). However, a group of strains, in particular those belonging to sequence type ST512 and producing KPC-3 carbapenemase as well as a clade of heterogeneous ST395 isolates expressing diverse carbapenemase types, exhibited a large degree of divergence in their ancestry, considering that the reference genome was ST11. Notably, two isolates (KP835819 and KP963668) belonging to the same high-risk ST (395) and harbouring a *bla*_OXA-48_ carbapenemase were clustered together, although the patients had no epidemiological link to each other and the isolates were collected more than six months apart, highlighting the benefits of cgMLST for the precise epidemiological tracking of CPE. The phylogenetic analysis of the *E. coli* isolates demonstrated a clear separation between strains producing VIM-1 and NDM-1 metallo-β-lactamases compared to the rest of the cohort. In addition, we observed the formation of phylogenetically related clades that were not apparent in the cgMLST analysis, particularly among strains harbouring *bla*_KPC_ and *bla*_OXA-484_ genes. The significance of these phylogenetic relationships must be determined by the epidemiological context surrounding the strains.

### 2.5. Plasmids

The plasmid analysis focused on plasmids associated with the presence of carbapenemase genes. Plasmids harbouring carbapenemase genes were identified in 93.4% of the isolates. For the *bla*_OXA-48_ gene, the predominant replicon type was IncL, with an average size of 50 Kb. Some 50% of samples were predicted to be conjugative, while the remaining 50% were non-mobilizable. In contrast, the *bla*_NDM-1_ gene was primarily associated with IncF replicon type, with an average size of 91.9 Kb. Some 78.5% of samples were conjugative, compared to 21.5% that were non-mobilizable. Overall, the most common plasmid replicon types were IncL and IncF, accounting for 60% of the isolates. The average size of plasmids harbouring carbapenemase genes was 81 Kb, ranging from 4.3 Kb (*bla*_OXA-48_) to 342.9 Kb (*bla*_NDM-1_), with 61.1% being conjugative, 4.7% being mobilizable, and 34.1% being non-mobilizable. Notably, multiple replicons were detected in 44.7% of isolates, and plasmids exceeding 80 Kb in size were consistently predicted to be mobile.

Table 3 summarizes the molecular characterization of the CPE grouped by species.

## 3. Discussion

From 2021 onward, we observed a consistent and progressive annual increase of approximately 20% in the total number of Enterobacteriaceae isolates in our hospital. This upward trend appeared to be strongly correlated with an increase in the volume of laboratory samples, particularly those sourced from primary care settings. This rise followed the reactivation of clinical services after the gradual de-escalation of COVID-19 restrictions. Notably, this surge in isolates was not observed in epidemiological samples collected between 2021 and 2022. During this period, most of the of epidemiological samples were collected from intensive care units that remained operational despite the pandemic-related restrictions, potentially limiting the increase. However, the significant escalation of Enterobacteriaceae isolates observed in 2023 can probably be attributed to the implementation and strengthening of surveillance programmes focused on MDR microorganisms in the community [25]. These programmes, introduced in 2022, played a crucial role in enhancing the detection and surveillance of MDR pathogens, thereby contributing to the increased identification of Enterobacteriaceae isolates in the following year.

In addition to the overall increase in Enterobacteriaceae isolates, the rise in CPE isolates was particularly pronounced, even after adjusting for the general increase in reported isolates over the same period. This significant increase highlights a notable shift in the local epidemiology of CPE, which may be attributed to several factors, including the introduction of strains from high-prevalence regions and/or the enhanced surveillance efforts focused on MDR microorganisms within hospital settings [26,27,28]. The species distribution of CPE isolates also differed significantly from global patterns, with *K. pneumoniae* being the predominant species, followed by *E. coli*, *Citrobacter* spp., and the *E. cloacae* complex. This deviation from global trends has been documented in other studies and may reflect species-specific traits that enhance the ability of these organisms to adapt to hospital environments [29,30]. In addition, the prevalence of carbapenemase production in these bacterial species may be facilitated by horizontal gene transfer mechanisms that promote the spread of resistance traits, further contributing to the evolving local epidemiology of CPE [31,32,33].

Although the majority of CPE isolates were obtained from epidemiological samples, it is noteworthy that several patients developed infections requiring antibiotic treatment for the same CPE strains they initially carried [34,35]. However, this clinical progression is not fully captured in the data, as only the first isolate received in the laboratory was included in the analysis. This methodological limitation means that subsequent infections or isolates from the same patients, which could reflect the development of clinically significant infections or changes in resistance profiles, were not accounted for. Consequently, the data may not fully reflect the dynamic nature of CPE colonization and infection in the patient population over time.

Consistent with findings from previous studies [36,37], a high level of resistance to the major classes of antibiotics commonly used in clinical practice was observed, particularly against third- and fourth-generation cephalosporins [38,39], fluoroquinolones [40], and aminoglycosides [41]. This pattern of multidrug resistance is likely attributed to several interrelated mechanisms, including the hydrolytic activity of carbapenemases that target and degrade other beta-lactams, the co-occurrence of genes encoding extended-spectrum beta-lactamases (ESBLs), and the presence of aminoglycoside-modifying enzymes on either the same or separate plasmids. Additionally, mutations in the *gyrA* and *parC* genes, which are selected based on prior fluoroquinolone exposure, contribute to resistance [42]. Interestingly, the relatively low rate of meropenem resistance observed in our CPE isolates may be explained by the high prevalence of class D carbapenemases. These enzymes show inconstant hydrolyzing patterns against meropenem, thus, exhibiting variable in-vitro results. Nevertheless, meropenem remains an unsuitable therapeutic option due to its reduced efficacy when facing these enzymes, despite showing low minimum inhibitory concentrations (MICs) [43]. Resistance to new-generation cephalosporins was also notable, with a particularly high rate of resistance to ceftolozane–tazobactam. This resistance is probably due to the limited activity of ceftolozane–tazobactam in isolates expressing carbapenemases, despite its potent anti-*Pseudomonas* activity [44]. In contrast, resistance to ceftazidime–avibactam was found to be consistent with the prevalence of metallo-beta-lactamases, as avibactam is effective against class A and D carbapenemases but lacks activity against class B enzymes [45]. In contrast to the findings reported by Lasarte-Monterrubio et al. in other isolates from Spanish hospitals, the high rates of cefiderocol-resistant strains observed in this study is remarkable. This is particularly noteworthy given that the intended use of cefiderocol, a recent addition to the antibacterial armamentarium, is to treat MDR bacterial infectios, including CPE. Mutations in genes that regulate iron transport and the expression of the carbapenemase VIM-1 have been proposed as factors contributing to cefiderocol resistance [46]. However, due to limited knowledge of resistance determinants and the need for strain-specific analysis, these factors were not included in the resistome analysis, although further investigation in future studies would be very valuable.

Despite fosfomycin and tigecycline exhibiting lower resistance rates compared to other antibiotics, more than one-third of isolates were non-susceptible to these agents. These antibiotics, along with colistimethate, have limited therapeutic efficacy and a higher risk of adverse effects [47], underscoring the need for tailored treatment strategies when managing CPE infections. Given these challenges, antibiotic susceptibility testing and carbapenemase characterization remain essential in order to guide appropriate and effective therapeutic interventions in the management of CPE infections.

The high prevalence of *K. pneumoniae*-producing OXA-48 carbapenemase observed in this study is consistent with both local and national epidemiological data, as reported by the Spanish Society of Infectious Diseases and Clinical Microbiology [48]. However, an emerging concern is the increasing diversity of carbapenemase types over time, including several alleles of *bla*_OXA-48_ as well as class A and B carbapenemases. This trend highlights the evolving complexity of resistance mechanisms within the local epidemiology of multidrug-resistant organisms. Notably, no single carbapenemase type was found to be exclusive to a particular species, suggesting the broader dissemination of these resistance traits across different bacterial species. This widespread distribution of carbapenemase-producing strains further emphasizes the need for active surveillance and comprehensive strategies to mitigate the spread of resistance.

The correlation between resistome analysis and antibiotic susceptibility profiles was highly consistent, with no major errors detected. However, it is crucial to note that the molecular detection of a resistance determinant (i.e., gene, allele, or mutation) does not always correspond to an observable phenotypic trait, as its expression may be influenced by gene expression. Similarly, a susceptible result for any individual carbapenem in the antibiotic susceptibility testing, based on clinical breakpoints, does not entirely exclude the presence of carbapenemases, as these enzymes may exhibit low hydrolytic activity or minimal expression at the time of testing. For certain antimicrobials, such as colistin, where resistance mechanisms involve multicomponent systems or remain incompletely characterized, establishing a robust correlation between molecular and phenotypic data is particularly challenging. Therefore, the cautious interpretation of resistome results is advised, with careful integration of data from antibiotic susceptibility testing whenever available.

The identification of key ST in *K. pneumoniae*, particularly high-risk clones such as ST147 and ST307, is consistent with previous reports of global outbreaks and the spread of carbapenemase-producing strains [49]. These STs are well documented in the spread of carbapenem resistance, highlighting the global nature of this problem [50,51]. Notably, two strains belonging to the hypervirulent ST23 clone were found to exhibit dual carbapenemase production, specifically *bla*_NDM-1_ and *bla*_OXA-48_, which are located on separate plasmids. This is the first reported instance of such an association in ST23, demonstrating the evolving complexity of resistance mechanisms and the need for enhanced genomic surveillance programmes in healthcare settings. The emergence of multidrug-resistant hypervirulent strains is of particular concern, as these isolates are associated with increased morbidity and mortality, representing a significant public health threat [52,53].

Interestingly, in contrast to the findings of other epidemiological surveillance studies on carbapenem-resistant *E. coli*, which commonly identify ST131, ST38, and ST405 as the predominant clones [54,55], our study revealed a different pattern. The isolates in this study exhibited marked heterogeneity in sequence types, with ST2659 being the most prevalent. Recently described in Algeria and Nepal [56,57], ST2659 is associated with *bla*_NDM_ alleles and belongs to clonal complex 38, a group that is linked to the global dissemination of carbapenemase-producing bacteria. The presence of ST2659 in this context further emphasizes the dynamic and evolving nature of carbapenemase-producing bacteria and the global spread of resistance determinants across different bacterial species and geographical regions.

Our study underlines the widespread presence of plasmids harbouring carbapenemase genes, with 93.4% of isolates carrying such plasmids. The absence of plasmid identification in the remaining isolates probably reflects the limitations of using only short-read sequencing for draft assembly. These plasmids may be split into multiple fragments in smaller contigs during assembly or may be present in low copy numbers within the isolates. It is plausible that the combination of current data with long-read sequencing could facilitate the accurate characterization of the mobile elements carrying the carbapenemase genes in these isolates. The predominant replicon types for *bla*_OXA-48_ and *bla*_NDM-1_ were IncL and IncF, respectively, and had different mobilization characteristics. Notably, 50% of *bla*_OXA-48_ associated plasmids were non-mobilizable, whereas a significant 78.5% of *bla*_NDM-1_ plasmids were conjugative. The identification of multiple replicons in nearly half of the cases highlights the complexity of carbapenemase gene transfer. The high mobility of plasmids larger than 80 Kb suggests the potential for rapid horizontal gene transfer, which may contribute to the spread of resistance in clinical settings.

## 4. Materials and Methods

### 4.1. Strains Included in the Study and Criteria for CPE Selection

All CPE isolated or submitted to HUMS from clinical or epidemiological samples between 1 January 2021 and 31 December 2023 were included in the analysis. Data on bacterial species, isolation date, the type of carbapenemase, and the antibiotic susceptibility profile were obtained from the Laboratory Information System (HORUS).

Inclusion criteria:Belonging to the Enterobacteriaceae family.The production of a carbapenemase confirmed by phenotypic or molecular methods.Having been isolated or submitted to HUMS within the specified time frame.

Exclusion criteria:The absence of a strain archive or viable culture.The contamination of the strain archive or culture.

Bacterial strain archives are stored in soy-tryptone glycerol 20% broth at −80 °C. In order to confirm strain identification, perform the macroscopical examination of culture purity, and establish biomass increases for WGS, subcultures on Columbia blood agar (Oxoid™ Thermo Fisher, Waltham, MA, USA), incubated at 35 °C for 24 h, were performed. Following manufacturer instructions, species-level identification was conducted using matrix-assisted laser desorption/ionization time-of-flight (MALDI-TOF) (Bruker Daltronics GmbH, Bremen, Germany) information from solid medium cultures, as well as routine laboratory procedures, accepting results with a score > 2.

Antibiotic susceptibility testing was performed by the broth microdilution method, using the MicroScan™ WalkAway semi-automated system (Beckman Coulter, Brea, CA, USA). The exception was cefiderocol testing, which was performed using the Kirby–Bauer method. Results were interpreted according to The European Committee on Antimicrobial Susceptibility Testing (EUCAST) guidelines [58]. Inconclusive results were confirmed via Kirby–Bauer or MIC test strip methods (Liofilchem® S.r.l., Roseto degli Abruzzi, Italy).

All CPE isolates were recovered from human clinical or epidemiological samples submitted to the HUMS Microbiology Laboratory as part of routine diagnostic procedures.

Screening for carbapenemase production was performed in all strains that presented an MIC of meropenem and or ertapenem ≥0.125 mg/L [59]. CPE in clinical samples were identified either by immunochromatographic assays NG-Test^®^ CARBA 5 (NG-Biotech Laboratories, Guipry-Messac, France) or genotypically via isothermal amplification techniques with Eazyplex^®^ (Amplex Diagnostics GmbH, Gars am Inn, Germany) or FilmArray^®^ (BioFire Diagnostics LLC, Salt Lake City, UT, USA), depending on the specific case and in accordance with laboratory protocols. All strains with a positive molecular result for carbapenemase detection were considered CPE, regardless of their minimum inhibitory concentration to carbapenems.

Epidemiological samples were initially screened for CPE using chromogenic selective media Brilliance™ CRE (Oxoid Limited, Basingstoke, UK), followed by a lateral flow assay NG-Test^®^ CARBA 5 (NG-Biotech Laboratories) or real-time polymerase reaction, with Xpert^®^ Carba-R (Cepheid, Sunnyvale, CA, USA) for confirmation of positive isolates.

### 4.2. WGS of CPE Isolates

All identified CPE isolates with pure and viable archive were candidates for WGS. Genomic DNA was obtained from 40 to 50 isolated colonies using a magnetic capture-based method with a MagCore^®^ system (RBC Bioscience, New Taipei City, Taiwan), following the manufacturer’s protocol, yielding 60 µL of eluate. The extracted genomic DNA was used to construct sequencing libraries using the Nextera XT™ Illumina^®^ kit (Illumina Inc., San Diego, CA, USA). Sample quantity and quality were verified at each step of the library preparation process using Qubit™ (Thermo Fisher Scientific, Inc.) fluorometric quantification and Bioanalyzer™ (Agilent Technologies, Inc, Santa Clara, CA, USA) analysis. Samples with a DNA concentration lower than 2 ng/uL or an insert size distribution outside 300 +/− 50 base pairs were discarded. Sequencing was performed on an Illumina^®^ MiSeq™ instrument using MiSeq V2 300 reagent cartridges (Illumina Inc., San Diego, CA, USA), employing a 150-base paired-end protocol and an expected average sequencing depth of greater than 30×. The library loading concentration target was 12.5 pM, with 5% phiX as the control quality.

De novo assembly was performed using Unicycler v0.5.0 [60], and the structural and functional quality of the assemblies was evaluated using QUAST v5.2.0 [61] and BUSCO v5.6.1 [62], respectively. Samples were rejected based on the following quality criteria: fewer than 800,000 reads with a quality score below 28 on the Phred scale, more than 10% Ns, an N50 of less than 30,000, or a genome size outside the range of 5.5 Mb ± 1.5 Mb. Contaminants in raw reads and assembled genomes were assessed using Mash v2.3 [63] and GUNC v1.0 [64], respectively. Finally, the assembly graphs were manually inspected using Bandage v0.8.1 [65]. Draft assemblies are under BioProject accession PRJNA1190923.

Species-level identification was verified using GAMBIT v1.0.1 [66] and the online PubMLST species identification service (https://pubmlst.org/species-id, accessed on 1 June 2024). Multi-locus sequence typing (MLST) was performed using mlst v2.23.0 [67], excluding for *K. pneumoniae*, which was analyzed separately. MLST and the annotation of antimicrobial resistance and virulence genes for *Klebsiella* sp. were performed with kleborate v2.4.1 [68]. Structural annotation was performed with prokka v1.14.6 [69], functional annotation was performed with sma3s v2 [70] (Uniref90 database), and resistance mechanisms were annotated using rgi v6.0.3 [71] (CARD v3.2.9 database) and abricate v1.0.1 [72,73] (ResFinder database). Resistance mechanisms with coverage > 80% and identity > 95% were considered.

Phylogenetic analysis was based on the alignment of single-nucleotide variants (SNVs) identified between the studied samples and the NCBI reference genome assemblies GCF_000240185.1 and GCF_000005845.2 for *K. pneumoniae* and for *E. coli*, respectively. Mapping, variant calling, and coreSNV calculation were performed using snippy v4.6.0 [74]. Recombination events were managed using the gubbins software v2.3 [75]. Phylogenetic inference and reconstruction were conducted using a maximum likelihood approach, with 1000 ultrafast bootstrap replicates and 1000 approximate likelihood ratio tests. The substitution models employed were TVM + F + ASC + R3 for *E. coli* and TVM + F + ASC + G4 for *K. pneumoniae*, which were implemented in iqtree v2.2.6 [76,77,78].

gcMLST was carried out for *E. coli* and *K. pneumoniae* using chewBBACA v3.3.1 [79] and the core genome schemes available from https://cgmlst.org/ncs, accessed on 1 July 2024. Training data files were generated using prodigal v2.6.3 [80], with the same reference sequences as those employed in the phylogenetic analysis. The results of the cgMLST analysis were then visualized by constructing a minimum spanning tree, created using the MSTreeV2 algorithm within the grapetree v2.2 software [81].

The plasmidome was reconstructed and assembled using the abricate v1.0.1 [72,82] software (PlasmidFinder database). The reconstruction and assembly of the plasmidome were conducted using MOB-suite v3.1.8 [83] and plasmidSPAdes v3.15.5 [84]. Further plasmidome annotation was performed using bakta v1.8.2 [85] (DB light v5.0 database) and rgi 6.0.3 [71], with visualization in the proksee software (https://proksee.ca/, accessed on 1 September 2024) [86].

Nucleotide identity analyses were carried out using fastANI v1.34 [87].

Inferential statistical analysis (Z-test to evaluate the difference between proportions) was performed using the prop_test function of the rstatix package in R v4.4.2.

## 5. Conclusions

The escalating prevalence of CPE in healthcare settings is an immediate and pressing public health challenge. This trend is particularly alarming given the limited therapeutic options available to treat CPE infections, which are often resistant to second- and third-line therapies. In light of our findings, the increase in the CPE prevalence, from 0.2% to 0.5%, of total Enterobacteriaceae isolates between 2021 and 2023 signals the need for ongoing and systematic surveillance to understand and address the dynamics of resistance. In this context, genomic characterization, in particular by WGS, is essential. This work also highlights the valuable information that WGS of CPE in a tertiary hospital setting could provide, including an in-depth understanding of the genetic determinants of resistance, high-resolution species identification, and the delineation of strain-level relationships, which in turn can be used by clinicians and public health authorities to guide therapy and take action to prevent or mitigate potential outbreaks.

## Figures and Tables

**Figure 1 antibiotics-14-00042-f001:**
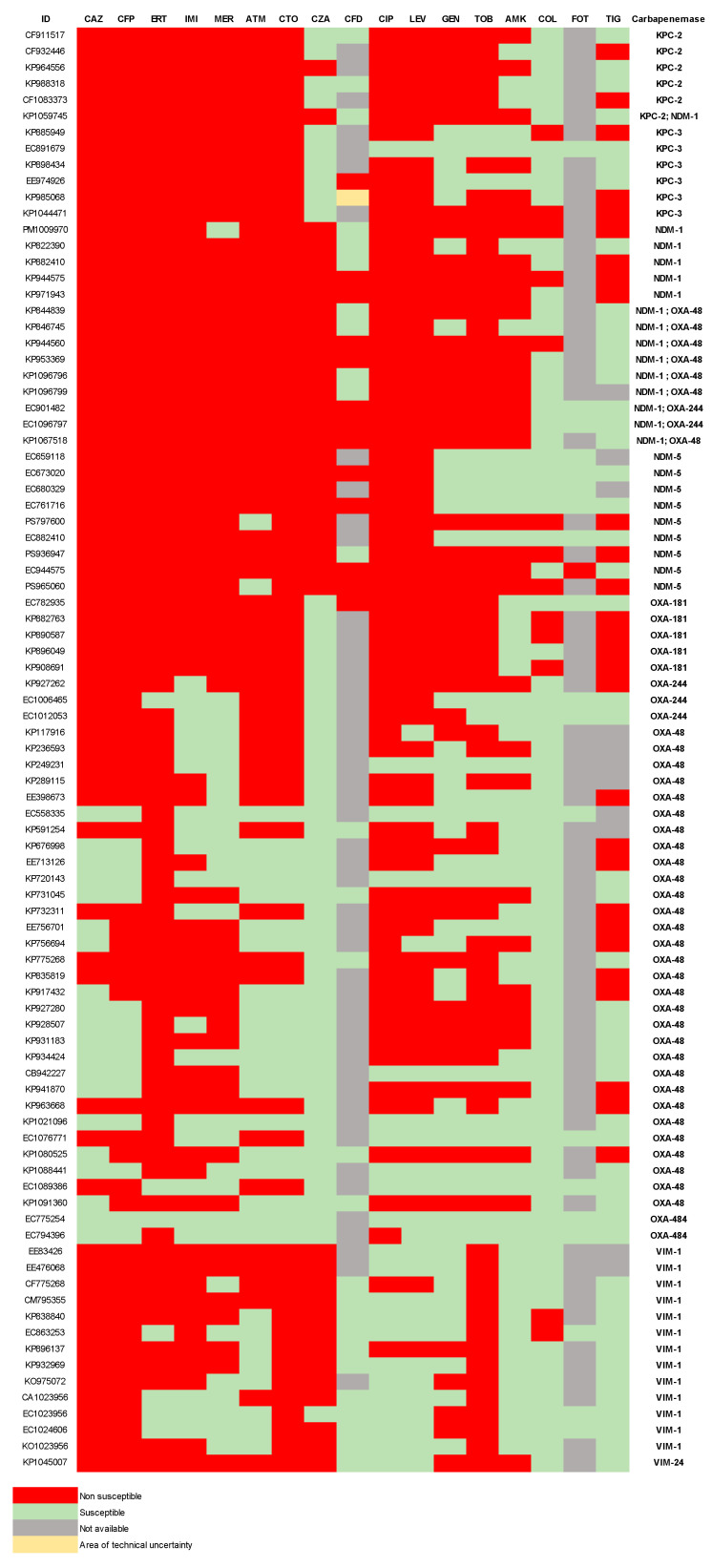
Heatmap of antibiotic susceptibility pattern by isolate. Heatmap showing susceptibility profiles and carbapenemase of each isolate. Green: susceptible; red: non-susceptible, grey: not available; yellow: area of technical uncertainty. ID: sample identification number. CAZ: ceftazidime; CFP: cefepime; ERT: ertapenem; IMI: imipenem; MER: meropenem; ATM: aztreonam; CTO: ceftolozane–tazobactam; CZA: ceftazidime–avibactam; CFD: cefiderocol; CIP: ciprofloxacin; LEV: levofloxacin; GEN: gentamicin; TOB: tobramycin; AMK: amikacin; COL: colistin; FOT: fosfomycin; TIG: tigecyclin.

**Figure 2 antibiotics-14-00042-f002:**
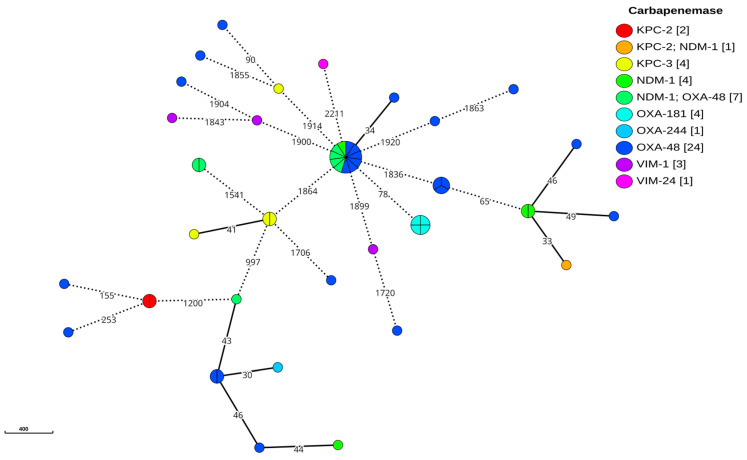
A MST of the *K. pneumoniae* cgMLST. Minimum spanning tree reconstructed from the allele matrix of the cgMLST of the *K. pneumoniae* species complex. Circles represent isolates, and are connected by solid lines with lengths proportional to the allelic distances between them. The allelic difference between each pair is indicated as an integer on the corresponding connecting line. Isolates with 15 or fewer allelic differences are grouped within the same circle, depicted as a pie chart, where the circle’s size reflects the number of isolates it contains. Dashed lines indicate allelic differences greater than 50. The number in brackets in the legend indicates the absolute frequency of the observation.

**Figure 3 antibiotics-14-00042-f003:**
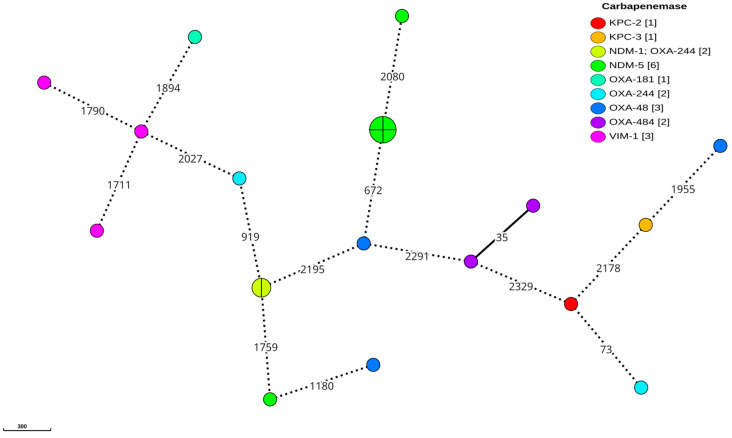
An MST of the *E. coli* cgMLST. Minimum spanning tree reconstructed from the allele matrix of the cgMLST of *E*. *coli*. Follows the same schematics as in Figure 2.

**Figure 4 antibiotics-14-00042-f004:**
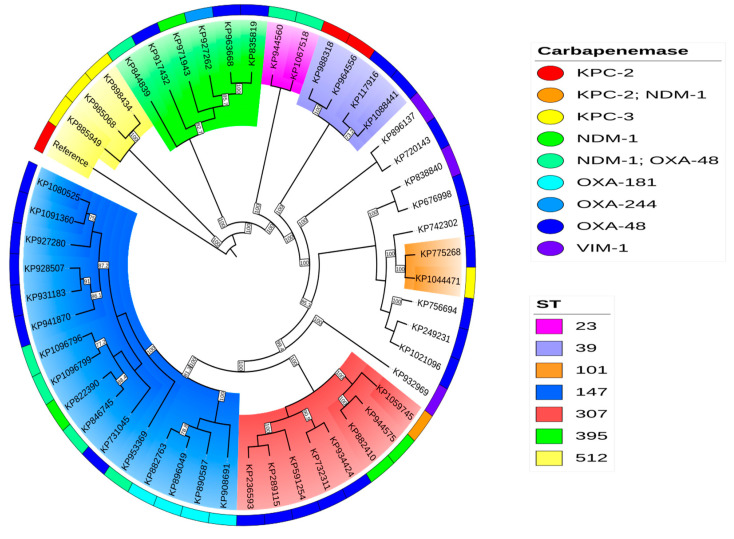
The phylogenetic analysis of carbapenem-producing *K. pneumoniae.* The phylogenetic analysis of *K. pneumoniae* sensu stricto. The cladogram depicts the relationships among isolates. Inner shading indicates the ST of each isolate. Isolates without shading represent unique STs. The outermost circle highlights the carbapenemase associated with each isolate. Bootstrap values exceeding 70 are displayed in white-background boxes on the respective branches.

**Figure 5 antibiotics-14-00042-f005:**
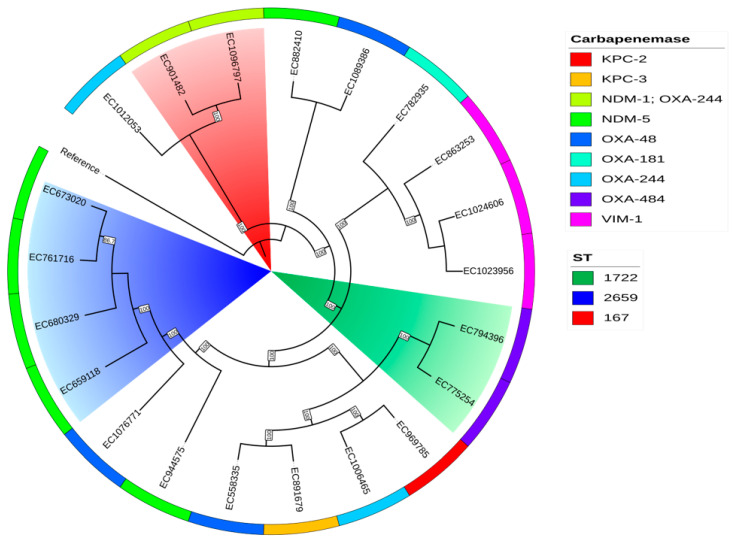
The phylogenetic analysis of carbapenem-producing *E. coli.* The phylogenetic analysis of *E. coli*. The cladogram depicts the relationships among isolates and follows the same schematics as in Figure 4.

**Table 1 antibiotics-14-00042-t001:** Distribution of Enterobacteriaceae isolation.

Species	2021	2021 Carba	2022	2022 Carba	2023	2023 Carba	Total
*E. coli*	6629	0	7590	7	8870	14	23,089
*K. pneumoniae*	1646	5	2102	14	2784	40	6532
*P. mirabilis*	649	0	741	0	891	1	2281
*E. cloacae*	424	3	542	2	579	3	1545
*K. oxytoca*	324	0	389	0	405	2	1118
*Citrobacter* spp.	280	5	357	7	455	7	1092
*S. marcescens*	205	0	217	0	246	1	668
*M. morganii*	177	0	261	0	284	0	722
*K. aerogenes*	142	0	162	0	179	0	483
*P. stuartii*	42	0	60	1	70	2	172
Others	110	0	126	0	207	0	443
Total	10,628	13	12,547	31	14,970	70	38,189

Total isolation of Enterobacteriaceae per year and the presence of carbapenemases. Carba: isolations with carbapenemases.

**Table 2 antibiotics-14-00042-t002:** The distribution of carbapenemases by species.

Carbapenemase	*K. pneumoniae* complex	*E. coli*	*E. cloacae* complex	*Citrobacter* spp.	*P. stuartii*	*K. oxytoca*	*P. mirabilis*	Total
KPC-2	2	1	0	3	0	0	0	6
KPC-3	4	0	2	0	0	0	0	6
NDM-1	4	0	0	0	0	0	1	5
NDM-5	0	6	0	0	3	0	0	9
VIM-1	3	3	2	3	0	2	0	13
VIM-24	1	0	0	0	0	0	0	1
OXA-48	24	3	3	1	0	0	0	31
OXA-181	4	1	0	0	0	0	0	5
OXA-244	1	2	0	0	0	0	0	3
OXA-484	0	2	0	0	0	0	0	2
NDM-1 + OXA-48	7	0	0	0	0	0	0	7
NDM-1 + OXA-244	0	2	0	0	0	0	0	2
NDM-1 + KPC-2	1	0	0	0	0	0	0	1
Total	51	20	7	7	3	2	1	91

**Table 3 antibiotics-14-00042-t003:** Genomic characterization summary of CPE.

CPE	Carbapenemases	ST	cgMLST Cluster	BLEE	AmpC	GyrA Mutation	*ParC* Mutation	Aminoglycosides Modifing Genes	Plasmid_Replicons (Carbapenemase)	Predicted_Mobility	Mean Plasmid Size (pb)
*K. pneumoniae*	OXA-48	13, 15, 39, 101, 147, 307, 346, 395, 405, 685, 4872	Yes	CTX-M-15	0	S83I, S83F, D87A	S80I	Several	IncL/M, IncR	Variable	46,443
	OXA-181	147	Yes	CTX-M-15	0	S83I	S80I	Several	rep_cluster_1195	Mainly non-mobilizable	40,801
	OXA-244	395	No	CTX-M-15	0	S83I	S80I	Several	IncL/M	Conjugative	87,763
	KPC-2	39	Yes	CTX-M-15	0	S83I, D87N	S80I	Several	IncFIB, IncFII, IncX3, ColRNAI_rep_cluster_1857	Mainly conjugative	94,778
	KPC-3	101, 512	Yes	0	0	S83I, S83Y, D87N	S80I	Several	IncFIB, IncFII, IncHI1B, IncR	Conjugative	63,166
	NDM-1	147, 307, 395	Yes	CTX-M-15	0	S83I	S80I	Several	IncFIB, IncFII, IncHI1B	Conjugative	278,576
	VIM-1	2, 685, 844, 387	No	0	DHA-1	0	0	Several	IncL/M	Conjugative	71,362
	KPC-2 + NDM-1	307	No	0	0	S83I	S80I	aph (3′)-VI only	IncFIB, IncHI1B, rep_cluster_1254	Conjugative	332,589
	NDM-1 + OXA-48	23, 147, 395	Yes	CTX-M-15	0	S83I	S80I	Several	IncFIB, IncFII, IncHI1B	Variable	101,820
*E. coli*	OXA-48	38, 127, 1598	No	CTX-M-15	0	S83L	0	Several	IncFIA, IncFII	non-mobilizable	21,541
	OXA-181	410	No	0	CMY-4	S83L, D87N	S80I	Several	IncX3	non-mobilizable	27,916
	OXA-244	44, 13,730	No	CTX-M-15	CMY-132	S83L, D87N	S80I	*aadA5* only	ND	ND	ND
	OXA-484	1722	No	0	0	0	0	*aadA2* only	rep_cluster_1195	non-mobilizable	14,615
	KPC-2	131	No	0	CMY-132	S83L	0	Several	IncFIB, IncFII, rep_cluster_2183	Conjugative	84,857
	KPC-3	135	No	0	0	0	0	*aph(3′)* only	IncX3	Conjugative	58,796
	NDM-5	46, 405, 2659	No	CTX-M-55	CMY-42	D87N, S83L	S80I	*aadA2* only	IncFIA, IncFIC, IncFIB	Mainly conjugative	105,572
	VIM-1	29, 327, 539	No	0	0	0	0	Several	IncL/M, IncL/M, IncI-gamma/K1	Variable	49,209
	NDM-1 + OXA-244	167	Yes	CTX-M-15	0	D87N, S83L	S80I	*aac(3)-IId* only	IncC, rep_cluster_1254	Conjugative	210,764
*Citrobacter* spp.	OXA-48	225	ND	0	CYM-101	0	0	0	IncL/M	Conjugative	61,961
	KPC-2	21,259	ND	CTX-M-15, CTX-M-9	CMY-106, CYM-159	S83I	S80I	Several	IncP, IncU	Mobilizable	29,296
	VIM-1	488,493,563	ND	CTX-M-9	CMY-2, CMY-48	0	0	Several	IncL/M, IncY	Mainly conjugative	57,471
*Enterobacter* spp.	OXA-48	13, 24	ND	CTX-M-15	ACT-1, MIR-5	S83I	0	0	IncL/M	Conjugative	64,843
	KPC-3	51	ND	0	ACT-40	0	0	Several	IncN	Conjugative	50,662
	VIM-1	108,198	ND	CTX-M-9	ACT-55	0	0	Several	IncL/M, IncHI2A	Conjugative	152,182
*Klebsiella* spp.	VIM-1	59	ND	CTX-M-9	0	S463A	0	Several	IncL/M, IncR	Conjugative	174,918
	VIM-24	4365	ND	CTX-M-9	0	S463A	0	Several	IncHI2A, rep_cluster_1088	Conjugative	289,821
*Providencia stuartii*	NDM-5	11, 23	ND	0	CMY-16	D87G	0	Several	IncC, rep_cluster_1254	Variable	78,040
*Proteus mirabilis*	NDM-1	446	ND	VEB-6	0	S463A	0	Several	ND	ND	ND

In double carbapenemase carriers, the plasmid description refers to the one harbouring the metallobetalactamase gene. BLEE: extended-spectrum betalactamase.

## Data Availability

Data were submitted to GenBank on 26 November 2024 under ID SUB14891856. BioProject accession: PRJNA1190923.

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
