# Peer review of "Genomic Characterization of Carbapenemase-Producing Enterobacteriaceae from Clinical and Epidemiological Human Samples"

_antibiotics, 2025, doi:10.3390/antibiotics14010042_

Round 1
Reviewer 1 Report
Comments and Suggestions for Authors
The authors should rearrange the flow of information and attend top minor grammatical and spelling errors.
Article title: Genomic characterization of carbapenemase-producing Entero- bacteriaceae using whole genome sequencing data
Review report
Overview
The reviewed manuscript tilted, Genomic characterization of carbapenemase-producing Entero- bacteriaceae using whole genome sequencing data, was well thought and the results and findings will be very useful in clinical management of resistant entro bacteria. The findings will further be useful in tracing AMR bacteria and genes. Improvement in antimicrobial stewardship will be another benefit from this study.
Abstract
The abstract was well written and summarised the entire work nicely from background through to conclusion. However, some details like materials and methods used in the study were not provided in the abstract.
Introduction
The introduction was well written and gave a good background information to the study aim. However, the following corrections and suggestions should be taken care of:
1. The authors should provide information on the generic mode of actions for the drugs (Carbapenems)
2. The authors should further describe carbapenemase enzymes and explain their interaction with the drugs.
Problem statement
1. The candidate should emphasise on the effect of fertilisers on bacterial diversity by citing known studies done elsewhere and establish the gap.
2. The candidate should state the role of microorganism in soil health.
3. In general, this section needs to be revised to fully reflect what the study intends to achieve.
Materials and Methods
This section explained well how the set objectives were to be archived. However, the authors should attend to the following;
1. Describe in full the sample collection plan, stating the inclusion and exclusion criteria.
2. Detailed explanation of the isolation and identification procedure (Media used etc)
3. Detailed explanation of the whole genome sequencing procedure (quality control aspects)
Results
The results were well presented, the tables and figures reflected the obtained data/information.
Discussion
The discussion section was well written; however, the authors should do well to review the grammar and spellings.
Conclusion
The conclusion was written well with major findings being teased out and well highlighted.
Comments on the Quality of English LanguageOnly minor spellings and grammatical which can easily be worked on
Author Response
Response to Reviewer 1 Comments
- Summary
Thank you very much for taking the time to review this manuscript. Please find the detailed responses below and the corresponding revisions/corrections highlighted/in track changes in the re-submitted files
- Point-by-point response to Comments and Suggestions for Authors
Comments 1: The abstract was well written and summarised the entire work nicely from background through to conclusion. However, some details like materials and methods used in the study were not provided in the abstract.
Response 1: Thank you for pointing this out. Abstract has been modified to expand in the materials and methods. Lines 21-27
Comments 2: The introduction was well written and gave a good background information to the study aim. However, the following corrections and suggestions should be taken care of:
- The authors should provide information on the generic mode of actions for the drugs (Carbapenems)
- The authors should further describe carbapenemase enzymes and explain their interaction with the drugs.
Response 2: Agree. We have, accordingly, revised the introduction and added a brief description of carbapenems and carbapenemases. Lines 55-69
Comments 3: Problem statement
- The candidate should emphasise on the effect of fertilisers on bacterial diversity by citing known studies done elsewhere and establish the gap.
- The candidate should state the role of microorganism in soil health.
- In general, this section needs to be revised to fully reflect what the study intends to achieve.
Response 3: While soil microorganisms and the use of fertilizers contribute to the generation, transmission, and reservoir of antibiotic resistance—particularly from a One Health perspective—we consider these aspects to fall beyond the scope of this study. Although it would be very interesting to addressing them in future research.
Comments 4: Materials and methods
This section explained well how the set objectives were to be archived. However, the authors should attend to the following;
- Describe in full the sample collection plan, stating the inclusion and exclusion criteria.
- Detailed explanation of the isolation and identification procedure (Media used etc)
- Detailed explanation of the whole genome sequencing procedure (quality control aspects)
Response 4: Agree. We have, accordingly, revised the materials and methods and expanded on the selection criteria, isolation and identification procedure and whole genome sequencing. Lines 497-522, and 555-568. Other aspects of raw data quality control are mentioned in lines 570-577.
- Response to Comments on the Quality of English Language
Point 1: Only minor spellings and grammatical which can easily be worked on
Response 1: A thoughtful review of the English language has been carried out and minor corrections has been made.
Please also find these comments in the attached file.

Reviewer 2 Report
Comments and Suggestions for Authors
The manuscript is well-written and easy to follow. The authors present a genomic characterization of carbapenemase-producing Enterobacteriaceae (CPE) using whole-genome sequencing (WGS) data.
Comments:
-
Title:
Consider revising the title to indicate that the samples were isolated from human specimens. -
Line 24:
The statement "Between 2021 and 2023, 0.4% of all isolates were CPE" would be clearer if the total number of isolates tested were specified. -
Tables and Figures:
The tables and figures are well-designed and effectively present the data. -
Phenotypic and Genotypic Resistance:
The authors performed both phenotypic and genotypic resistance testing. Please discuss how the isolates’ resistance profiles align or differ between these methods for each antimicrobial tested.
Author Response
Response to Reviewer 2 Comments
- Summary
Thank you very much for taking the time to review this manuscript. Please find the detailed responses below and the corresponding revisions/corrections highlighted/in track changes in the re-submitted files
- Point-by-point response to Comments and Suggestions for Authors
Comments 1: Consider revising the title to indicate that the samples were isolated from human specimens
Response 1: Thank you for pointing this out. Title has been modified to explicitly stand sample provenance.
Comments 2: The statement "Between 2021 and 2023, 0.4% of all isolates were CPE" would be clearer if the total number of isolates tested were specified.
Response 2: Agree. We have, accordingly, modified that statement in the abstract to make it clearer. Lines 28-29
Comments 3: The authors performed both phenotypic and genotypic resistance testing. Please discuss how the isolates’ resistance profiles align or differ between these methods for each antimicrobial tested.
Response 3: Agree. We have, accordingly, expanded the discussion to address this matter. Lines 459-471.
Please also find these comments in the attached file.

Reviewer 3 Report
Comments and Suggestions for Authors
1. MST figures need better resolution. Especially, the numbers are not clearly visible. The legends needs to be made bigger for more clarity.
2. Proofreading required for minor spelling and grammatical mistakes throughout the manuscript.
Author Response
Response to Reviewer 3 Comments
- Summary
Thank you very much for taking the time to review this manuscript. Please find the detailed responses below and the corresponding revisions/corrections highlighted/in track changes in the re-submitted files
- Point-by-point response to Comments and Suggestions for Authors
Comments 1: MST figures need better resolution. Especially, the numbers are not clearly visible. The legends needs to be made bigger for more clarity.
Response 1: Thank you for pointing this out. Legends in all figures have been enlarged for clarity. MST allele numbers are now presented with enhanced contrast for better visibility. All figures are in high resolution, but unfortunately, some quality loss may occur during the import process into the Word file. I have attached the original images for your review.
Comments 2: Proofreading required for minor spelling and grammatical mistakes throughout the manuscript.
Response 2: A thoughtful review of the English language has been carried out and minor corrections has been made.
Please also find these comments in the attached file.
